# Judgments of learning in bilinguals: Does studying in a L2 hinder learning monitoring?

**Marta Reyes**[1]*, **Mª Julia Morales**[2], **Mª Teresa Bajo**[1]

1 Mind, Brain, and Behavior Research Center (CIMCYC), University of Granada, Granada, Spain,
2 Universidad Loyola Andalucía, Sevilla, Spain

* mreyessanchez@ugr.es

## Abstract

Nowadays, use of a second language (L2) has taken a central role in daily activities. There are numerous contexts in which people have to process information, acquire new knowledge, or make decisions via a second language. For example, in academia and higher education, English is commonly used as the language of instruction and communication even though English might not be students' native or first language (L1) and they might not be proficient in it. Such students may face different challenges when studying and learning in L2 relative to contexts in which they study and learn in their L1, and this may affect their metamemory strategies. However, little is yet known about whether metamemory processes undergo significant changes when learning is carried out in L2. The aim of the present study was to investigate the possible consequences on learning derived from studying materials in L2 and, more specifically, to explore whether the interplay between monitoring and control (metamemory processes) changes as a function of the language involved. In three experiments, we explored whether font type (Experiment 1), concreteness (Experiment 2), and relatedness (Experiment 3) affected judgments of learning (JOLs) and memory performance in both L1 and L2. JOLs are considered the result of metacognitive strategies involved in the monitoring of learning and have been reported to vary with the difficulty of the material. The results of this study showed that people were able to monitor their learning in both L1 and L2, even though they judged L2 learning as more difficult than L1. Interestingly, self-perceived difficulty did not hinder learning, and people recognized L2 materials as well or better than L1 materials. We suggest that this might be an example of a desirable difficulty for memory.

## Introduction

Over the past decades, use of a second language (L2) has become part of daily life for many. There are numerous contexts in which bilingual people with very different linguistic profiles process information, acquire new knowledge, or make decisions via a second language. For example, in academia and higher education, English is commonly used as the language of instruction and communication [1] even though English might not be students' native or first language (L1) and they might not be proficient in it. Such students may face different

**Data Availability Statement:** Data have been uploaded to the Open Science Framework (OSF) public repository as separate .csv files for each experiment, as well as codebooks as .txt files with

the explanation of each variable. Please, find the data in the following OSF identifier: https://osf.io/e7qyn/?view_only=1e2034b6d38641bab4a460172a530997.

**Funding:** This research was supported by the doctoral research grant FPU18/01675 to Marta Reyes; by grants from the Spanish Ministry of Economy and Competitiveness (PGC2018-093786-B-I00 30B51801); from the Spanish Ministry of Science and Innovation (PID2021-127728NB-I00); and from Junta de Andalucía (A-CTS-111-UGR18/B-CTS-384-UGR20/P20_00107) to Teresa Bajo.

**Competing interests:** The authors have declared that no competing interests exist.

challenges when studying and learning in L2 relative to contexts in which they study and learn in their L1. Despite the fact that bilingual instruction contexts tend to be the norm and that L2 use has grown over the years, research on whether the processes underlying learning undergo significant changes when the learning context is not L1 is still scarce [for an attempt at exploring the mechanisms underlying non-native primary students' underachievement, see [2]. The focus of the present study is to investigate the factors that regulate metacognitive processes of learning in L2 contexts.

Even for highly proficient bilinguals, working in L2 can be cognitively challenging [3–5]. A large number of studies have established that the two languages are co-activated within the bilingual brain when producing or comprehending in both written and spoken modalities and even in contexts where only one of the languages is involved [6–10]. If the alternative language remains active when using the context-appropriate language, additional cognitive resources are recruited to control the interference and actively select the desired language [11–14]. In this respect, brain imaging studies have revealed that neural bases of bilingual language control share brain networks with processes that enable domain-general cognitive control [e.g., 15]. According to the adaptive control hypothesis [16], bilingual people need to recruit control and meta-control processes so as to maintain the language goal, monitor the conflict, and suppress possible interference from language co-activation. The type and form of control exerted depends on the context and on the type of language experience of the bilingual person [14].

Language control might be especially costly for unbalanced bilinguals, since the interference from the dominant L1 to the less-dominant L2 has been shown to be greater than the other way around [e.g., 13, 17, 18]. Moreover, unbalanced and late bilinguals rely primarily on transfer from L1 to L2. Semantic representations are weaker in L2, and some concepts are activated through L1-L2 translation [19]. Moreover, some studies have shown that L2 processing takes 20% more time, word recognition is slower, and a smaller amount of information is processed simultaneously compared to L1 [20]. All this suggests that L2 processing is more challenging and might take place within a presumably overloaded cognitive system [3, 21–23]. The question is then whether this cognitive overload has consequences for learning strategies and cognitive resources allocated when processing, studying, and acquiring new information in L2.

From a learning perspective, metacognition is a key function that serves a self-regulatory purpose whereby the brain monitors the learning conditions and regulates the resources and processes devoted to learning. Metacognition is one of the key components of self-regulated learning [24–26] and is involved in the development of successful learning strategies linked to academic achievement [25, 27, 28].

According to Nelson and Narens' [29] classic model, there are two mechanisms underlying metacognitive strategies in the learning process: *monitoring* and *control. Monitoring* refers to the online supervision and assessment of the effectiveness of cognitive resources. Monitoring processes, such as judgements regarding the ease/difficulty of the material and tasks or the level of learning achieved after studying, are crucial to unfolding *control* processes, such as selecting a strategy, regulating the cognitive resources devoted to the task, and adjusting them when necessary [29, 30]. Thus, a task's perceived difficulty, uncertainty, complexity, or novelty, for example, may serve as cues to trigger *control* strategies. These two functions of metacognition (monitoring and control) are inextricably connected and, in turn, have consequences on memory. Some theories assume that accurate monitoring leads to appropriate regulation for the benefit of learning and memory performance [31–34].

The interplay between monitoring and control is assumed to be effortful and cognitively demanding [35]. For these two processes to occur, sufficient cognitive resources must be available and executive control must be engaged, as the flow of bottom-up (*monitoring*) and top-down (*control*) processes is simultaneous [29]. In support of this assumption, research with

younger and older people suggests that metacognitive processes recruit cognitive resources. For example, Stine-Morrow et al. [36] showed that learning in an older group was reduced relative to a group of younger participants when memory monitoring was required, suggesting that metacognitive monitoring might compromise performance in the age group whose executive control skills might be in decline. Similarly, Tauber and Witherby [37] showed that, unlike for younger adults, instructions to use metacognitive strategies did not improve memory performance in older adults, suggesting that age-related deficits might make it difficult for older participants to implement these strategies. Similarly, neuropsychological studies provide evidence that the neural correlates of metamemory are driven by frontal lobes [38]. Finally, a review of brain imaging studies reveals that midfrontal and frontoparietal areas are involved in metacognition [39, 40], which suggests that executive functioning is involved in metacognition [41].

*Judgments of learning* (JOLs) are one of the procedures used to assess monitoring processes. In a classical task, participants are asked to rate (usually with a percentage) the likelihood of remembering in the future the learning material they have just studied. They can be based on a full-list evaluation (e.g., lists of words, texts) or an item-specific assessment (e.g., single words, pictures, etc.) [42]. JOLs are inferential in nature and combine information from different sources. These include inherent features of the material, such as perceptual characteristics (e.g., size and clarity), association strength, word frequency, concreteness, or relatedness; conditions of encoding and testing (time frame, test format, presentation rate, retention interval, etc.); and one's own memorial experience of the material [43].

Therefore, people's JOLs have been shown to be sensitive to variations in the to-be-studied material, such as perceptual, lexical, and semantic features or the degree of coherence and elaboration. For instance, participants tend to judge that their memory will be better for easy-to-read items in contrast to difficult-to-read items [44], for concrete in contrast to abstract words [45–47], and for related pairs [48, 49] and semantically related word lists [50] in contrast to unrelated words, even under conditions of divided attention while studying [51]. Whether the effects of JOLs on memory performance rely on the ease of processing at encoding, on general beliefs, or on a combination of both factors is still debated, but overall, the evidence suggests that people use cues at different processing levels to assess the difficulty of the learning process.

Generally, JOLs tend to be quite accurate in predicting recall performance (e.g., concrete and related words usually receive higher JOLs and are usually better remembered than abstract and unrelated words [46, 49, 52]). However, some studies have demonstrated dissociations between JOLs and memory, with participants exhibiting underconfidence or overconfidence regarding their predictions about their success in remembering the targets [e.g., 53]. According to the cue-utilization approach [43], JOLs reflect inferential processes based on cues provided by the materials and tasks, and mispredictions may arise because the cues used by the learner are not diagnostic, informative, or related to actual memory performance [e.g., see 54] for mixed results in the fluency effect of JOLs and memory].

Taken together, previous research indicates that people are sensitive to different characteristics when judging the likelihood of remembering the studied material. However, given that monitoring is cognitively effortful, it is also possible that when studying in L2, people have less cognitive resources available to devote to such metacognitive and learning processes. Whether the language of study plays a part in the monitoring process and whether it interacts with other cues is yet to be known.

## Present studies

In three experiments, we aimed to investigate the consequences of studying in L1 or L2 on the interplay between memory monitoring and control. We explored the effects of manipulating

font type, concreteness, and relatedness on JOLs and recognition to discover to what extent unbalanced bilinguals can monitor and control their memory both in L1 and L2. Previous research exploring the consequences of studying in L1 vs. L2 on memory have found different effects depending on the type of test. For example, Vander Beken et al. [55, 56] found that essay questions hindered performance in L2 presumably due to difficulties in writing production while no differences between L1 and L2 performance were found with True/False recognition questions when students studied short expository texts.

Additionally, Mizrahi et al [57] predicted that bilinguals would be disadvantaged in free recall, but advantaged in recognition in the nondominant language. In this vein, previous studies confirmed that bilinguals recalled fewer items, exhibited worse memory for item order, and weaker primacy effects in the nondominant than the dominant language [58–60]. In addition, two more studies reported better recognition memory for words in bilinguals' nondominant compared to their dominant language [61, 62]. Thus, the process of writing seems to be more complex and challenging in L2 due to linguistics aspects and proficiency. In order to avoid confounding effects with writing complexity, in our experiments we used recognition tests to assess memory performance.

In the experiments, participants studied lists of words and provided JOLs for each of the study words (Experiments 1 and 2) or for short study lists (Experiment 3). All participants performed the tasks in both L1 and L2, with language being blocked and counterbalanced. Across experiments, we varied the features of the materials that could be considered cues for metacognitive assessment. Experiment 1 concerned perceptual features (font type); Experiment 2 concerned lexical-semantic features (concreteness); and Experiment 3 concerned semantic-relational features (relatedness among words in a short list). Thus, our manipulation involved different types of processing that may interact with the language of study in different ways [3, 4, 13, 20, 63]. Note that these manipulations are not equally predictive of learning success since, intrinsically, perceptual manipulation does not necessarily imply increased difficulty of the material [64] neither when encoding nor during retrieval. However, people do encode and retrieve concrete and categorically related words better than abstract and unrelated words, which makes them more memorable [65, 66]. Hence, manipulations of the different features of the to-be-studied materials might reveal interesting interactions between language and monitoring.

Overall, our hypothesis was that the extra cognitive demands involved in L2 processing relative to L1 processing may reduce metacognitive processing and monitoring as they both (L2 processing and monitoring) require cognitive control. We expected to observe less accurate use of possible cues for monitoring in L2 than in L1, meaning that manipulations to increase or decrease the intrinsic difficulty of the material–such as concreteness or relatedness of the to-be-studied material–might not be detected when the task is performed in L2. In addition, we wanted to assess whether participants adjust their overall perception of learning to the context provided by the language. Since each block in the experiments defined a language context (either L1 or L2), it was possible to assess if the participants perceived learning success differently depending on the language and adjusted their JOLs consequently. Overall, we expected that participants considered learning in L1 to be easier and more successful than learning in L2, with higher JOLs in the L1 context than in the L2 context.

## Experiment 1: Easy-to-read vs. difficult-to-read font type

Experiment 1 represents an initial attempt to explore how the language of study interacts with a perceptual manipulation of the to-be-studied material. With this purpose, we examined the effect of font type on JOLs and memory when the study materials were presented in L1 or in

L2. Interestingly, the evidence for the effect of perceptual manipulations on JOLs and memory performance is mixed and seems to depend on specific conditions [see 54 for a review]. Over-all, results suggest that the use of perceptual fluency as a cue for JOLs is evident when there are no other cues available (e.g., item relatedness, study time) and when the fluency is manipulated within participants [67]. Regarding memory performance, results are also mixed. Some studies indicate that perceptual disfluency may function as a desirable difficulty since it provides a metacognitive cue for "difficulty," leading to more effortful processing and, in turn, to better performance [44, 68, 69]. Yet, some other studies have concluded that perceptual disfluency does not affect memory [44, 64, 70]. The question then is whether perceptual fluency might be boosted as a cue for learning monitoring when more effortful processing is dedicated to study due to the L2 condition. In Experiment 1, we manipulated font type (easy- vs. difficult-to-read) and language of study (L1 vs. L2) within participants, but language was blocked and counterbalanced so that, within each block, font type was the main cue on which to base the JOLs whereas the language acted as the contextual setting of learning (i.e., within each block words appeared in the same language). As mentioned, we were interested not only in whether there were differences in the way participants assessed their learning within each language context, but also in whether they adjusted their JOLs as a function of the language context (e.g., higher JOLs for L1 than L2).

## Methods

**Participants.** We conducted a power analysis using G*Power [71] to determine the required sample size. We calculated it considering a mixed factor analysis of variance (ANOVA) with language and font type as repeated measure variables, and order of the language block as a between-participants variable. We estimated a required sample size of 28 participants, assuming a small-to-moderate effect size (partial eta squared of 0.05) to observe significant (α = 0.05) effects at 0.8 power. This estimation applies for all three experiments, as they are all of similar design and characteristics. We recruited some more participants in order to ensure a representative sample after removing those who did not perform the task correctly: participants who did not vary JOLs across items, had a low hit and/or high false alarm proportion (d-prime < 0.5) in the recognition test [72], and/or had a rate of fast anticipatory responses (<300ms) of over 10% [73] were excluded from the study. Participants had normal or corrected-to-normal vision and reported no neurological damage or other health problems. Participants gave informed consent before participating in the experiment. The experiment was carried out following the Declaration of Helsinki [74]. The protocol was approved by the institutional Ethical Committee of the University of Granada (857/CEIH/2019) and the Universidad Loyola Andalucía (201222 CE20371).

Thirty-seven psychology students from the University of Granada participated in Experiment 1. We removed from all analyses a participant who rate every item at the maximum possible value. We therefore had a total sample of 36 (18–40 years old, *M* = 23.67, *SE* = 5.34). Participants were tested in person and individually in the laboratory and received course credit as compensation.

We recruited non-balanced Spanish-English bilinguals who started acquiring English as their L2 during late childhood (*M* = 7.90, *SE* = 2.86). They were moderately proficient in English, as proven by subjective [Language Background Questionnaire, LEAP-Q; 75] and objective (MELICET Adapted Test, Michigan English Language Institute College Entrance Test, and verbal fluency task) language measures. S1 Table in S1 File shows descriptive statistics for the participants in this and all other experiments.

**Materials and procedure.** The experimental session lasted approximately 100 minutes. The main task consisted of a JOL task with a study phase, a distractor task, and a recognition

test. Additionally, participants completed other cognitive tasks, a metacognitive questionnaire regarding the strategies used when studying the words, a language background and sociodemographic questionnaire [LEAP-Q; 75], a verbal fluency test in L1 and L2, and the MELICET Adapted Test. We presented stimuli and collected data for all tasks with E-Prime Professional 2.0 software [76].

The JOL task was modeled after Halamish [77]. This paradigm included two consecutive blocks with identical procedure with the exception of the language in which words were written (Spanish–L1 vs. English–L2). The assignment of L1 and L2 to the blocks was counterbalanced across participants as a between-subject factor. For each block, participants studied a list of words for a later recognition test and were informed that the words would be presented in two different font types. For each block, words could appear in either an easy-to-read (Arial 18 points black color, RGB (Decimal) 0, 0, 0 and RGB (Hex) 0x0, 0x0, 0x0) or *difficult-to-read* font (Monotype Corsiva 18 points silver color, RGB (Decimal) 192, 192, 192 and RGB (Hex) 0xC0, 0xC0, 0xC0), following the procedure established by previous studies [78–80]. Each study phase lasted eight minutes, and the recognition test took approximately four minutes. During the study phase, words were presented one at a time in the middle of the computer screen. For each trial, a fixation point appeared for 500 ms; then, a slide with the study word remained for 5,000 ms. Immediately after the presentation of each word, participants gave a judgment of learning (JOL). They predicted the likelihood of remembering it on a 0–100 scale (0: not likely at all, 100: very likely). They typed in the JOL using a regular computer keyboard. This screen advanced automatically after the prescribed time (4,000 ms) or when the participant pressed ENTER.

For each block, the to-be-studied list comprised 44 words, with the first and last two words serving as the primacy and recency buffers and the remaining 40 as targets. Language was blocked such that all words within each study/recognition block appeared either in L1 or L2. Within each language block (L1 or L2), participants studied 40 words (after removing primacy and recency buffers), half of them in an easy-to-read font and half of them in a difficult-to-read font type, which was counterbalanced across participants. For assignment to the easy/difficult-to-read conditions, we created two lists (list A and list B) of 20 words in each language. The assignment of each list to the easy-to-read or to the *difficult-to-read* font was counterbalanced across participants (word lists for this and the next two experiments can be found in S2 File).

We selected English and Spanish words from the CELEX English Corpus [81] and the LEX-ESP database [82] and used the N-Watch [83] and BuscaPalabras programs [84], respectively, to compute and control for psycholinguistic indices. Within and between lists and languages, words were matched for estimated frequency (L1-List A: $M = 2.2$, $SD = 0.4$; L1-List B: $M = 2.2$, $SD = 0.5$; L2-List A: $M = 2.3$, $SD = 0.4$; L2-List B: $M = 2.3$, $SD = 0.4$) and number of letters (L1-List A: $M = 4.8$, $SD = 1.0$; L1-List B: $M = 4.8$, $SD = 0.9$; L2-List A: $M = 5.1$, $SD = 1.1$; L2-List B: $M = 4.8$, $SD = 0.8$). Within the blocks, words were presented in a pseudo-random order, with the restriction that no more than three items from the same font type appeared consecutively.

In between the study phase and the recognition test of each block, participants did a distractor task for 10 minutes. We chose a short version of the AX- Continuous Performance Task [AX- CPT; 85], which is a cognitive control task with minimum verbal load.

With regard to the recognition test, studied words (excluding the primacy and recency buffers) appeared along with 40 new words in a random order. Studied and new words were two independent sets that remained constant for all participants, but they were matched for mean estimated frequency (L1: $M = 2.1$, $SD = 0.3$; L2: $M = 2.3$, $SD = 0.3$), and mean number of letters (L1: $M = 4.7$, $SD = 1.2$; L2: $M = 4.4$, $SD = 0.8$), so that any possible effect that may arise would

not be explained by those psycholinguistic parameters [see 44, 86, 87 for a similar procedure].
First, a blank slide was displayed for 100 ms. Then, the target stimulus remained on the screen
for 3,000 ms or until the participant gave a response. For each word, participants indicated
whether it had appeared in the study phase by pressing a 'YES' or 'NO' key. The assignment of
the keys (Z and M) to the correct responses ('YES' and 'NO') was counterbalanced between
subjects and kept constant across tasks.

**Results.** We performed 2 x 2 x 2 (language x font type x block order) mixed-factor ANO-
VAs for JOLs in the study phase and accuracy (d-prime) in the recognition test. Language (L1
vs. L2) and font type (easy-to-read vs. *difficult-to-read*) were within-subject factors, and block
order (L1-first vs. L2-first) was a between-subject factor. We included block order in the analy-
ses because participants' calibration and expectations when performing the task may vary as a
function of whether the first block was performed in L1 or L2. For all analyses, the alpha level
was set to 0.05, and we corrected by Bonferroni for multiple comparisons. Effect sizes are
reported in terms of partial eta squared ($\eta_p^2$) for ANOVAs and Cohen's *d* for t-tests.

We also conducted the same analyses but including MELICET scores as a covariate (ANCO-
VAs), which yielded identical results. MELICET did not interact with language $F(1, 33) = 0.749$,
$p = .393$, $\eta_p^2 = .002$, or font type $F(1, 33) = 0.05$, $p = .829$, $\eta_p^2 = .000$ for JOLs or recognition
task–language $F(1, 30) = 0.29$, $p = .593$, $\eta_p^2 = .001$ and font type effect $F(1, 30) = 0.02$, $p = .887$,
$\eta_p^2 = .000$). Therefore, for the sake of simplicity, we report ANOVAs outcomes in the main text.

We removed three duplicate items in the L1 block, which resulted in 76 valid trials (37 stud-
ied and 39 new items) in the L1 block and 80 valid trials (40 studied and 40 new items) in the
L2 block. Measures were adjusted for the total number of valid items in each language block.

**Study phase (JOLs).** To evaluate the effect of language and font type on the magnitude of
JOLs, we computed the mean across participants after removing trials with blank responses
(0.63%) and trials with responses over 100 (0.77%, presumably due to typography errors, as
participants were instructed to rate their JOLs on a 0–100 scale by key-pressing a value).

We found no significant main effects of language ($F(1, 34) = 0.63$, $p = .432$, $\eta_p^2 = .02$), font
type ($F(1, 34) = 2.18$, p = .149, $\eta_p^2 = .06$), or block order ($F(1, 34) = 0.01$, $p = .922$, $\eta_p^2 = .000$).
We observed a significant interaction between language and block order ($F(1, 34) = 9.47$, $p = .004$, $\eta_p^2 = .22$). Post-hoc comparisons revealed that when L2 was studied first, JOLs for L2
items were lower ($M = 57.8$, $SE = 4.7$) than for L1 items ($M = 66.8$, $SE = 3.9$), although these
differences were marginal ($t(34) = 2.74$, $p = .059$). In contrast, when participants started with
the L1 block, they rated comparably the probability of remembering items in both languages
($M = 60.2$, $SE = 3.9$ and $M = 65.5$, $SE = 4.7$ for L1 and L2 items, respectively; $t(34) = -1.61$, $p = .696$). No other interaction was significant: font type did not interact with block order ($F(1, 34) = 0.61$, $p = .44$, $\eta_p^2 = .02$) or with language ($F(1, 34) = 0.39$, $p = .535$, $\eta_p^2 = .01$). Block order
and language did not interact neither (reporter) and the three-way interaction between font
type, language, and block order was not significant ($F(1, 34) = 0.47$, $p = .499$, $\eta_p^2 = .01$). Over-
all, it seemed that font type did not have an effect in either L1 or L2.

**Recognition test (accuracy).** Following the procedure of Undorf and Zander [88], we
removed from the analysis trials with a reaction time shorter than 300 ms (0.14% of the total
number of trials). We calculated d-prime as a sensitivity index on the basis of hits and false
alarms. Greater d-prime indicates better discrimination between studied and new items. We
followed Hautus [see 89] and the 1/(2N) rule to apply corrections for extreme false-alarm or
hit proportions (p = 0 or p = 1). Due to a technical error, we did not record data from the rec-
ognition test for the first three participants. Therefore, we analyzed data from 33 participants
in this measure.

The analysis showed that d-prime did not differ across conditions. Neither language ($F(1, 31) = .61$, $p = .441$, $\eta_p^2 = .02$), font type ($F(1, 31) = 0.70$, $p = .41$, $\eta_p^2 = .02$), nor block order ($F$

**Table 1. JOLs and d-prime across conditions.**

| Language | Block order | Easy-to-read | | Difficult-to-read | |
|---|---|---|---|---|---|
| | | JOL | d-prime | JOL | d-prime |
| L1 | L1-first | 60.0 (4.0) | 2.9 (0.2) | 60.4 (4.1) | 2.9 (0.2) |
| | L2-first | 67.9 (4.0) | 2.4 (0.2) | 65.8 (4.1) | 2.5 (0.2) |
| L2 | L1-first | 66.4 (4.6) | 2.8 (0.2) | 64.7 (4.7) | 2.7 (0.2) |
| | L2-first | 58.8 (4.6) | 2.5 (0.2) | 56.8 (4.7) | 2.4 (0.2) |

$(1, 31) = 3.55$, $p = .069$, $\eta_p^2 = .10$) reached significance. None of the interactions were significant. Block order did not interact with language ($F(1, 31) = .15$, $p = .701$, $\eta_p^2 = .01$) or font type ($F(1, 31) = .10$, $p = .751$, $\eta_p^2 = .003$). Neither did font type interact with language ($F(1, 31) = 1.85$, $p = .184$, $\eta_p^2 = .06$). The three-way interaction between the factors did not reach significance ($F(1, 31) = 0.72$, $p = .40$, $\eta_p^2 = .02$). Participants recognized items similarly across conditions. Additional information about estimated means (and standard deviations) for hits, false alarms, misses and correct rejections by language, font type and block order can be found in S1 File. See Table 1 for estimated means and standard deviations for JOLs and d-prime.

**Goodman–Kruskal gamma correlation.** We used a Goodman–Kruskal (GK) gamma correlation [90] as a nonparametric measure of the association between JOL and subsequent recognition. This analysis permitted us to examine participants' metamemory accuracy–resolution–across conditions. We calculated one gamma correlation for each participant in each of the four conditions of interest (L1 easy-to-read, L1 difficult-to-read, L2 easy-to-read, L2 difficult-to-read). We then ran mixed-factor ANOVAs to examine whether the GK gamma correlations differed across conditions, including block order as a between-subject variable. Note that the degrees of freedom may differ from the previous analyses because the correlation cannot be computed when there is not enough variance in participants' responses [91].

We found a marginal effect of font type ($F(1, 16) = 4.18$, $p = .058$, $\eta_p^2 = .207$), with the difficult-to-read font type ($M = 0.3$, $SE = 0.1$), having better resolution than the easy-to-read font type ($M = 0.2$, $SE = 0.1$). This was mediated by a marginal interaction between language and font type ($F(1, 16) = 4.18$, $p = .058$, $\eta_p^2 = .207$). We observed a tendency towards having better resolution for the difficult materials in L2, whereas easy-to-read and difficult-to-read materials did not differ in L1 (see Table 2). The main effect of block order was not significant ($F(1, 16) = 0.04$, $p = .85$, $\eta_p^2 = .002$). Neither was any of the other interactions (all $ps > .05$).

With regard to the learning strategies used in the study phase, in this experiment participants reported grouping words by their semantic meaning (86.1%), creating mental images (69.4%) and rehearsal of words (52.8%) as the strategies most used.

## Discussion

In Experiment 1, we were interested in two possible effects. First, we wanted to observe if a perceptual cue, such as font type, produced different JOLs and recognition accuracy and if

**Table 2. Goodman–Kruskal gamma correlations across conditions.**

| Language | Block order | Easy-to-read | Difficult-to-read |
|---|---|---|---|
| | | JOL | JOL |
| L1 | L1-first | 0.2 (0.2) | 0.2 (0.2) |
| | L2-first | 0.3 (0.1) | 0.2 (0.2) |
| L2 | L1-first | 0.0 (0.2) | 0.3 (0.2) |
| | L2-first | 0.1 (0.1) | 0.3 (0.2) |

they differed as a function of the language context in which the task was performed. Second, we were interested in assessing if the linguistic context (L1 or L2) had an effect on the overall perceived learning difficulty of the task.

Regarding font type, we did not find an effect of font type on JOLs in either language (L1 or L2). Participants predicted similar memory performance for words in a difficult-to-read and easy-to-read font. Correspondingly, recognition accuracy was similar for both font conditions. This pattern of results is in line with Magreehan et al. [67], who did not find an effect of font type when other cues were available. It can be argued that with our design and materials, font-type was the only cue available for the participants since language was blocked. However, participants had been fully informed of the procedure, and they knew from the beginning that they were going to study words in two languages and that within each language block, words could appear in two different font types. They were instructed to judge their learning based on the difficulty perceived with all the information available, which includes font type and language. In fact, even when participants reported similar JOLs in both font type conditions, gamma correlations indicated that they seemed to actually monitor the difficult material better than the easy material, especially in L2, where the correlation between JOL and recognition seems to suggest better adjustment between perceived degree of learning and actual recognition performance. That is, when participants performed the task in L2, JOLs for remembered difficult-to-read items were higher than for unremembered ones in L2, suggesting they monitored difficult materials better. However, this interaction was only marginal, and it should be considered with caution.

Interestingly, the language effect was dependent on the order in which the languages were presented. Thus, participants' JOLs increased for L1 when L2 was presented first, whereas when the L1 block preceded the L2 block, differences in the perceived difficulty of the language did not reach significance. Although block order did not modulate the GK correlations, still this effect might be due to the possible anchor point for further comparison provided by the first block. It is possible that participants were cautious in judging their degree of learning during the initial L2 block and increased their JOLs when confronted with the following, easier L1 block. This increase in perceived learning for the second block was not evident when the second block was L2. Note, however, that the greater perceived difficulty of L2 when L2 was presented first did not correspond with performance in the recognition test, since recognition did not vary with language or language order.

In sum, the results of Experiment 1 suggest that when font type was manipulated, there were very small variations in JOLs and recognition accuracy, and the latter did not vary with language.

## Experiment 2: Concrete vs. abstract words

In Experiment 2, we introduced a lexical-semantic manipulation by including concrete and abstract words in the study list. Concrete words have richer and more interconnected semantic representations than abstract words, and have been shown to enhance item memory [92–94]. Concreteness effect have been demonstrated not only in memory but also in JOLs [46, 52]. Concrete words are more easily processed, and this encoding fluency serves as a cue for meta-cognitive judgements, such as JOLs [46].

We expected that this manipulation might interact with language, since conceptual processing has been shown to differ across languages [95–97]. Thus, associations between words and their meanings have been shown to be weaker in L2 than in L1 [19], especially for unbalanced bilinguals (see the Revised Hierarchical Model by Kroll and Stewart [19]), and this may have an effect when the cue for learning monitoring also involves conceptual processing.

Thus, we expected that, consistent with previous studies, participants might give higher JOLs to concrete relative to abstract words in L1 [52]. However, for L2, concreteness might not be so evident in JOLs and/or memory, as participants might not be able to monitor and detect the difficulty and to adjust to it. In addition, we wanted to explore if, similar to Experiment 1, we would observe a language-by-block interaction, indicating that participants adjust their JOLs depending on the anchor point provided by the language of the first block.

## Methods

**Participants.** Participants were selected following the same criteria and procedure described in Experiment 1. Thirty-nine psychology students from the University of Granada participated in this experiment. We removed a participant who gave JOL values of 100% to all items, suggesting he/she did not perform the main task correctly, resulting in a final sample of 38 (18–31 years old, $M = 21.37$, $SD = 2.86$). Participants were tested individually in an online experiment and received course credit as compensation. In this experiment, we included self-reported measures for L1 in LEAP-Q. Comparisons of all self-reported measures and of the verbal fluency test results showed that participants were unbalanced and significantly more fluent in L1 than in L2. All *p* values were below .05. See S1 Table in S1 File for descriptive statistics.

**Materials and procedure.** Participants were tested in a single online remote session that lasted approximately 120 minutes. We programmed, presented the stimuli and collected data for all tasks with Gorilla Experiment Builder, an online platform [98]. Participants accessed the experiment individually and on their own. They were forced to full-screen presentations so as to prevent them from opening other windows in the computer while doing the tasks. Recent research supports the validity and precision of experiments run online [98–100].

The procedure was similar to that of Experiment 1, since the same cognitive and linguistics tasks were administered, although they were administered through an online platform in this case. In addition, for the memory and JOL tasks, word concreteness was manipulated. In this experiment, for the JOL task, participants responded by using the mouse to move a handle slider to the desired number and pressed the spacebar to continue to the next word. As in Experiment 1, the language of the study phase and test (L1 and L2) was blocked and counterbalanced. The study lists were composed of 44 nouns (4 primacy and recency buffers and 40 targets), and the subsequent recognition tasks included 80 words (40 targets and 40 new words). Half of the study and recognition words were concrete (concreteness for L1: $M = 5.8$, $SD = 0.5$; L2: $M = 4.6$, $SD = 0.4$) and half were abstract (L1: $M = 3.8$, $SD = 0.7$; L2: $M = 2.6$, $SD = 0.7$), and they were presented in random order. We selected words from Brysbaert et al. [101] and translated them to obtain words in Spanish–L1. Across participants, the L1-L2 versions of the words were counterbalanced in such a way that words that appeared in L1 for one participant would not appear in the L2 block. All selected words were composed between 3 and 7 letters and medium frequency. Within languages, concrete and abstract studied and new words were matched in estimated frequency (L1: $M = 2.0$, $SD = 0.3$; L2: $M = 2.2$, $SD = 0.3$) and numbers of letters (L1: $M = 5.3$, $SD = 1.2$; L2: $M = 5.0$, $SD = 1.2$). Note that we used two language specific norms to select the words. Concreteness ratings for English words were based on Brysbaert et al., [101] using a 5-point scale for English, whereas values for the Spanish words were based on LEXESP [82] using a 7-point scale. Thus, the descriptive statistics are in different scales. However, the criteria to consider a word abstract or concrete was equivalent for both data set. We calculated the concreteness mean for each language, and words with ratings above the means in both languages were considered concrete, while words with values below the means were considered abstract. See S5 Table in S1 File for concreteness ratings of each list.

## Results

As in Experiment 1, we report mixed-factor ANOVAs for JOLs and d-prime in the recognition test. Language (Spanish–L1 vs. English–L2) and concreteness (abstract vs. concrete words) were within-subject factors, and block order (L1-first vs. L2-first) was a between-subject factor. As in Experiment 1, we conducted the same analyses but including MELICET scores as a covariate (ANCOVAs), which yielded identical results. Scores in the MELICET did not interact with language $F(1, 35) = 2.67$, $p = .112$, $\eta_p^2 = .071$, or concreteness $F(1, 35) = 0.66$, $p = .420$, $\eta_p^2 = .019$ for JOLs or for recognition–language $F(1, 33) = 0.11$, $p = .739$, $\eta_p^2 = .003$, and concreteness effect $F(1, 33) = 3.56$, $p = .068$, $\eta_p^2 = .097$).

We removed two items that were erroneously duplicated in both blocks (Spanish–L1 and English–L2). This resulted in 79 valid trials (40 studied and 39 new items) in both blocks. Measures were adjusted for the total number of valid items in each language block. We removed from all analyses a participant who did not vary the JOLs across the items, suggesting he/she was not performing the task correctly.

**Study phase (JOLs).** The results showed no significant main effects of language ($F(1, 36) = 0.17$, $p = .684$, $\eta_p^2 = .005$) or block order ($F(1, 36) = 0.15$, $p = .904$, $\eta_p^2 = .000$). We observed a significant main effect for concreteness ($F(1, 36) = 18.34$, $p < .001$, $\eta_p^2 = .34$) such that concrete words received higher JOLs ($M = 59.4$, $SE = 3.0$) than abstract words ($M = 55.0$, $SE = 2.8$). There was a marginal interaction between language and block order ($F(1, 36) = 4.00$, $p = .053$, $\eta_p^2 = .10$). Follow-up tests revealed non-significant effects. Neither was there a significant difference between languages depending on whether the L1 block (L1: $M = 54.5$, $SE = 4.3$; L2: $M = 59.1$, $SE = 4.5$) or L2 block (L1: $M = 59.1$, $SE = 4.1$; L2: $M = 56.0$ $SE = 4.3$) was placed first. However, we observed a tendency of crossover effects such that L1 had lower JOLs when the L1 block was placed first and L2 received lower JOLs when the L2 block was placed first. No other interaction was significant: concreteness did not interact with block order ($F(1, 36) = 0.44$, $p = .513$, $\eta_p^2 = .012$), or with language ($F(1, 36) = 2.08$, $p = .158$, $\eta_p^2 = .055$). The three-way interaction was not significant ($F(1, 36) = 0.00$, $p = .931$, $\eta_p^2 = .000$).

**Recognition test (accuracy).** Following the procedure in Experiment 1, we filtered out trials with fast responses (<300ms, 0.21%). We removed a participant who had a reaction time below 300ms in more than 10% of trials [73] and another participant whose d-prime was below 0.5 (low hit or high false alarm proportion) [72].

For d-prime, the analysis showed a significant main effect of concreteness ($F(1, 34) = 29.95$, $p < .001$, $\eta_p^2 = .468$). Participants recognized concrete words ($M = 2.6$, $SE = .12$) better than abstract words ($M = 2.3$, $SE = .11$). In addition, there was a significant language-by-block order interaction effect ($F(1, 34) = 6.96$, $p = .013$, $\eta_p^2 = .170$). Thus, words in L2 were better recognized ($M = 2.7$, $SE = 0.17$) than words in L1 ($M = 2.3$, $SE = 0.17$), but only when the L2 block was placed first ($t(34) = -3.03$, $p = .028$). When participants started with the L1 block, they recognized items in both languages similarly (L1: $M = 2.5$, $SE = 0.18$; L2: $M = 2.4$, $SE = 0.18$, $t(34) = 0.76$, $p = .100$). We did not observe other significant main effects or interactions ($p > .1$ for all). Additional information about estimated means (and standard deviations) for hits, false alarms, misses and correct rejections by language, concreteness and block order can be found in S1 File. See Table 3 for estimated means and standard deviations for JOLs and d-prime.

**Goodman–Kruskal gamma correlation.** In the mixed-factor ANOVA, we found no significant effect of block order ($F(1, 19) = 0.19$, $p = .665$, $\eta_p^2 = .01$) (L1-first: $M = 0.2$, $SE = 0.1$; L2-first: $M = 0.2$, $SE = 0.1$), language ($F(1, 19) = 0.60$, $p = .449$, $\eta_p^2 = .031$) (L1: $M = 0.2$, $SE = 0.1$; L2: $M = 0.3$, $SE = 0.1$), or concreteness ($F(1, 16) = 0.14$, $p = .709$, $\eta_p^2 = .008$) (concrete: $M = 0.2$, $SE = 0.1$; abstract: $M = 0.2$, $SE = 0.1$). None of the interactions were significant.

**Table 3. JOLs and d-prime across conditions.**

| Language | Block order | Concrete | | Abstract | |
|---|---|---|---|---|---|
| | | JOL | d-prime | JOL | d-prime |
| L1 | L1-first | 56.9 (4.6) | 2.7 (0.2) | 52.1 (4.2) | 2.4 (0.2) |
| | L2-first | 62.1 (4.4) | 2.5 (0.2) | 56.1 (4.0) | 2.1 (0.2) |
| L2 | L1-first | 60.5 (4.7) | 2.5 (0.2) | 57.8 (4.5) | 2.3 (0.2) |
| | L2-first | 58.1 (4.4) | 2.8 (0.2) | 53.9 (4.3) | 2.6 (0.2) |

With regard to the learning strategies used in the study phase, in this experiment participants reported creating mental images (68.8%), words rehearsal (59.4%), grouping words by their semantic meaning (59.4%), and relating words to personal experiences (56.3%) as the strategies most used in the study phase.

## Discussion

The results of Experiment 2 replicated the concreteness effect that has been commonly reported in previous metamemory studies; that is, concrete words produced larger JOLs and better recognition rates than abstract words [52, 102]. Consistent with previous studies [103], despite judging abstract words as more difficult than concrete words, participants did not seem to allocate sufficient resources to compensate and achieve the same recognition as concrete words. Importantly, these effects were evident independent of whether participants performed the tasks in L1 or L2.

As for the language cue, there was a tendency to judge L1 words as better learned than L2 words. Interestingly, when the L2 block was placed first, participants seemed to compensate and achieve better memory for L2 words than for L1 words. The fact that L2 words were recognized better than L1 words is in line with previous studies reporting better recognition in the less-fluent language [61, 62]. It is possible that participants devoted more resources to what they slightly perceived as slightly more difficult (L2) and ultimately achieved better learning. Interestingly, this compensatory effect was evident in Experiment 2 but not in Experiment 1. Materials in Experiment 2 included concrete and abstract words, which might have induced semantic processing, and this may be the factor underlying the effect. In Experiment 3, we assessed this explanation by introducing a relational-semantic dimension which might make these compensatory effects even more evident.

## Experiment 3: Words grouped into semantic categories vs. unrelated words

In Experiment 3, we introduced a semantic manipulation, namely the degree of within-list semantic organization, which requires relational processing and semantic integration [66]. Relational processing and organization are among the most efficient processes for learning [104]. Organization involves awareness of the sematic relations of the material during encoding and the use of this organization at retrieval. Hence, organization as a learning strategy involves a high degree of metacognitive processing at both encoding (assessment of possible word relations) and retrieval (controlled organizational strategies). Previous research on metamemory has reported on the effect of relatedness in memory and JOLs. People systematically give higher JOLs, and indeed recall and recognize related information (pairs or lists), better than information that is unrelated [49, 105, 106]. However, research across older and younger participants has also shown that organization at encoding and retrieval is often impaired in older participants [66, 107–111], suggesting that the use of these strategies involves the

engagement of fully intact control processes. Previous research on L2 language processing has also shown that processes such as inferencing or mental model updating during text comprehension are impaired when the texts are presented in L2 relative to L1 [3]. Hence, it is possible that engagement of language control during L2 processing might also reduce the use of costly encoding and retrieval strategies relative to L1 processing.

Consistent with previous literature, we expected to find an effect of relatedness in both JOLs and recognition in L1. However, as this material requires deeper associative processing, we also expected that it would be affected by the possibly costlier monitoring and memory processes in L2.

## Methods

**Participants.** Forty-two psychology students from the University of Granada (45.2%) and Universidad Loyola Andalucía (54.8%) participated in this experiment. They were recruited and selected following the same procedure and criteria as in Experiments 1 and 2. We removed a participant who recorded the default JOL value of 50% for all items, suggesting he/she did not correctly perform the main task. This resulted in a final sample of 41 (18–29 years old, $M = 20.54$, $SD = 2.41$). Participants were tested individually in two remote sessions and received course credit as compensation. Comparisons for all self-reported measures and for the verbal fluency test showed that participants were unbalanced and significantly more fluent in L1 than in L2. All $p$ values were below .05. See S1 Table in S1 File for descriptive statistics.

**Materials and procedure.** The experiment consisted of two online sessions, each of which lasted approximately 60 minutes. The procedure was similar to that of Experiment 2, with the same cognitive and linguistics tasks programmed and administered with the same experiment builder [98]. However, the procedure for the memory task differed in the materials used and the moment when JOLs were solicited. In this case, participants studied 10 short lists of six words for a later recognition test and gave a JOL after the study phase for each list. We used an adapted procedure, modeled after Matvey et al. [50]. Lists comprised either words grouped into a semantic category (e.g., musical instruments: horn, bass, drum, keyboard, harp, saxophone) or unrelated words (e.g., hole, blind, tower, kingdom, wheel, bishop). Nevertheless, unlike Matvey et al. [50], participants in our study gave JOLs after studying each list instead of giving JOLs after each target word. Note that the relatedness manipulation affects the complete list (related word lists vs unrelated-word lists), differently from experiments 1 and 2 were the manipulation affected specific words within the list (e.g., concrete words vs abstract words), and therefore, in this case, we could assess the difficulty of the list as a whole.

Within each list, words were presented randomly, one at a time, in the middle of the computer screen. Within each session, participants completed the JOL and recognition task in the two language blocks (L1 vs. L2). Similar to previous experiments, the order of the language blocks was counterbalanced across participants.

Semantic categories were selected from Van Overschelde et al. [112] for English words and Marful et al. [113] for Spanish words. We excluded English and Spanish cognates and filtered categories with less than six exemplars. Unrelated and new words were randomly selected from Brysbaert et al. [101]. Studied semantic-category words and unrelated studied and new words within and between languages were matched for estimated frequency and number of letters (see S6 Table in S1 File for frequency and number of letters descriptive of each study list).

Materials for the study phase consisted of 20 lists of six words. For half of the lists, words belonged to the same semantic categories, whereas for the other half, the lists were composed of unrelated words. We randomly assigned five semantic-category lists and five unrelated lists to the Spanish–L1 and English–L2 block. List assignments were counterbalanced across

participants, and participants were randomly assigned to one of the counterbalanced conditions. The lists within a counterbalanced condition were pseudo-randomly presented with the restriction that no more than two consecutive lists belonged to the related or unrelated condition. For the recognition task, participants were presented with all studied words and 60 unrelated new words, for a total of 120 words. Note that new words in the recognition task were always unrelated because given the restriction in the selection procedure, there were not enough categories to be used as new-distractor words. However, this was true for the two language conditions, and therefore the critical between-language comparison was fully controlled. Words appeared randomly one by one in the center of the screen, regardless of the condition (words grouped into semantic related categories and unrelated words).

## Results

As in Experiments 1 and 2, we introduced the JOLs from the study phase and d-prime for recognition into ANOVAs with language (Spanish–L1 vs. English–L2) and relatedness (related list vs. unrelated list) as within-subject factors and block order (L1-first vs. L2-first) as a between-subject factor. As in Experiment 1 and 2, we conducted the analyses but also including MELICET scores as a covariate (ANCOVAs), which yielded identical results. Scores in the MELICET did not interact with language $F(1, 38) = 3.30$, $p = .007$, $\eta_p^2 = .080$, or type of list $F(1, 38) = 0.001$, $p = .972$, $\eta_p^2 = .000$ when considering both the JOLs and recognition test–language $F(1, 31) = 0.42$, $p = .520$, $\eta_p^2 = .013$, and type of list $F(1, 31) = 0.001$, $p = .970$, $\eta_p^2 = .000$.

Following the same exclusion criteria as in the previous experiments, we removed one participant from analyses.

**Study phase (JOLs).** The results of the ANOVA yielded a significant main effect of language ($F(1,39) = 9.29$, $p = .004$, $\eta_p^2 = .192$), with L1 lists ($M = 64.4$, $SE = 2.5$) receiving higher JOLs than L2 lists ($M = 59.3$, $SE = 2.1$). We also observed a main effect of relatedness ($F(1,39) = 84.10$, $p < .001$, $\eta_p^2 = .683$), such that related lists ($M = 71.6$, $SE = 2.4$) received higher JOLs than unrelated lists ($M = 52.1$, $SE = 2.4$). The main effect of block order was significant ($F(1,39) = 5.41$, $p = .025$, $\eta_p^2 = .122$). JOLs tended to be higher when the L2 block was placed first ($M = 66.9$, $SE = 3.12$) compared to JOLs when L1 was first ($M = 56.8$, $SE = 3.1$). There were no significant interactions ($p > .1$ for all).

**Recognition test (accuracy).** In this task, we excluded two participants for having a fast response (<300ms) in more than 10% of trials [73] and five participants with d-prime below 0.5 [72], resulting in a sample of 34 participants. We filtered out 0.16% trials with reaction times shorter than 300 ms.

We found a language effect ($F(1, 32) = 5.51$, $p = .025$, $\eta_p^2 = .147$), with L2 lists ($M = 2.4$, $SE = 0.15$) being recalled better than L1 lists ($M = 2.2$, $SE = 0.13$). This effect was mediated by a significant interaction effect of language and block order ($F(1, 32) = 7.29$, $p = .011$, $\eta_p^2 = .186$). Thus, participants recognized L2 words ($M = 2.7$, $SE = 0.21$) better than L1 words ($M = 2.1$, $SE = 0.18$) only when the L2 block was placed first. We also observed a type-of-list effect ($F(1, 32) = 35.56$, $p < .001$, $\eta_p^2 = .526$). Words grouped into semantic categories ($M = 2.5$, $SE = 0.13$) were better recognized than unrelated words ($M = 1.9$, $SE = 0.13$) regardless of language or block order. There were no other significant main effects of block order or interactions ($p > .1$ for all). Additional information about estimated means (and standard deviations) for hits, false alarms, misses and correct rejections by language, type of list and block order can be found in S1 File. See Table 4 for estimated means and standard deviations for JOLs and d-prime.

**Goodman–Kruskal gamma correlation.** In order to calculate the Goodman–Kruskal gamma value for each subject and condition, we correlated the JOLs with the proportion of

**Table 4. JOLs and d-prime across conditions.**

| Language | Block order | Semantic category | | Unrelated words | |
|---|---|---|---|---|---|
| | | JOL | d-prime | JOL | d-prime |
| L1 | L1-first | 69.9 (4.0) | 2.3 (0.2) | 50.0 (4.0) | 1.8 (0.2) |
| | L2-first | 77.4 (4.1) | 2.4 (0.2) | 60.2 (4.1) | 1.8 (0.2) |
| L2 | L1-first | 63.1 (3.2) | 2.3 (0.2) | 44.1 (3.4) | 1.7 (0.2) |
| | L2-first | 75.9 (3.3) | 2.9 (0.2) | 54.2 (3.5) | 2.6 (0.2) |

words correctly recognized in each list. The mixed-factor ANOVAs revealed no significant difference across conditions. There was no significant main effect of block order ($F(1, 25) = 2.48$, $p = .128$, $\eta_p^2 = 0.09$) (L1-first: $M = -0.2$, $SE = 0.1$; L2-first: $M = 0.0$, $SE = 0.1$), language ($F(1, 25) = 0.95$, $p = .339$, $\eta_p^2 = 0.037$) (L1: $M = -0.2$, $SE = 0.1$; L2: $M = -0.1$, $SE = 0.1$), or relatedness ($F(1, 25) = 0.02$, $p = .903$, $\eta_p^2 = 0.001$) (words grouped into semantic categories: $M = -0.1$, $SE = 0.1$; unrelated words: $M = -0.1$, $SE = 0.1$). None of the interactions were significant ($p > .1$ for all).

With regard to the learning strategies used in the study phase, in this experiment participants reported words rehearsal (76%), grouping words by their semantic meaning (68.3%), and creating mental images (56.1%) as the strategies most used in the study phase.

## Discussion

We replicated the relatedness effect in both JOLs and memory. As expected, JOLs and recognition were higher for related lists of words in L1, and this was also true for L2-word lists. Moreover, JOLs were lower for L2 than for L1 lists, indicating that with these materials, participants found the L2 block more difficult than the L1 block. Interestingly, they recognized L2 words better, especially when the L2 block was placed first. Thus, as in Experiment 2, we found an L2 recognition advantage [61, 62]. The perceived difficulty might have triggered some kind of deeper processing to compensate and achieve successful learning.

## General discussion

The goal of the present study was to explore the consequences of studying in L2 contexts on the metacognitive processes required for successful learning. In three experiments, we found that learning in L2 did not fully compromise the monitoring of learning. Participants could judge the materials accurately in both L1 and L2, and they only found difficulty in L2 blocks under some circumstances and with certain materials. More interestingly, language of study seems to have a differential effect on monitoring depending on the features of the material of study.

The three experiments differ in the type of cues provided by the materials to guide learning monitoring: perceptual (font type, Experiment 1), lexical-semantic (concreteness, Experiment 2), or semantic-relational (relatedness, Experiment 3), both in L1 and L2. While we did not find differences in JOLs due to perceptual cues (people did not find the *difficult-to-read* font less likely to remember than the easy-to-read font in Experiment 1), participants did report differential JOLs due to the lexical and semantic cues (giving lower JOLs to abstract and unrelated words than to concrete and related words in Experiments 2 and 3, respectively). These results are in line with previous research—concreteness and relatedness effects consistently appear in the JOL literature [50, 52], whereas font type effects are not so consistently found [67], and they do not usually appear in recognition [64, 114, 115]. More importantly, language did not impede monitoring under any of our manipulations; learning monitoring was similarly performed in L1

and in L2. The fact that our late, unbalanced bilinguals were equally able to assess their degree of learning for features such as concreteness and relatedness in L1 and L2 has clear practical implications, since it suggests that, at least for simple materials, learning monitoring and control are not impaired for bilinguals with medium-high level of proficiency; therefore, L2 instruction and materials can be safely introduced in educational and academic settings.

Regarding the language context for the study, participants judged L2 blocks as more difficult than L1 blocks. However, this effect varied across experiments, such that in Experiments 1 and 2, where JOLs involved single words, this pattern was only evident when the L2 block was placed first. In contrast, in Experiment 3, where JOLs involved judging short lists, the language effect appeared independently of list or block order. These language effects were independent of the type of font (Experiment 1), the degree of concreteness (Experiment 2), or the degree of relation between the words in the list (Experiment 3). It is possible that when considering complete lists, learners might be required not only to activate mental representations of the word but also to access representations of associated words. Memory links between words and conceptual representation have been shown to be stronger in L1 than in L2 [19], and therefore, words might have stronger links in L1 than in L2, and this, in turn, might manifest in differential judgements for L1 and L2. Different relational representations for L1 and L2 would be independent of whether the L2 block was presented first or second, and therefore, the language effect in Experiment 3 is independent of the block order. In contrast, when the task involved single words and no further relational processing was required, JOLs depended on the calibration between the two blocks; thus, participants considered L1 to be easier (increased their JOLs) after first experiencing the more difficult L2 language condition. Hence, it seems that participants used the first block as a baseline for comparison and considered L1 to be easier when contrasted with the more difficult L2 block. The reason why this contrast effect was only obtained when participants were first presented with L2 is not evident and might be due to a number of reasons (e.g., more interference from L1 that balances out the possible benefits of greater effort for L2); therefore, this should be a subject of further research. However, in line with many previous bilingual studies [14, 116, 117], these results provide evidence that the obtained L2 effects are dependent on the context in which L2 learning is achieved.

These context effects were also evident when looking at memory performance; L2 materials were better recognized than L1 materials when the L2 block was performed first. This interaction effect between language and block order in recognition is also in line with previous studies reporting that bilinguals performed worse in their L1 if they were first tested in their L2 [e.g., 118–120]. However, this interaction was only present in Experiments 2 and 3, where our manipulations induced semantic-type processing (concreteness and relatedness), and it was not preset when the manipulation induced superficial processing (font type). The bilingual L2 advantage in recognition memory has previously been attributed to the greater episodic distinctiveness and lower familiarity of L2 words and to greater demand for cognitive resources [61, 62]. Thus, in our experiments, it might be the case that control regulatory mechanisms engaged more cognitive resources to the task even though the monitoring process had not clearly identified a potential deficit in learning. Thus, participants might have paid more attention to L2 words because they believed they would be less likely to be remembered, which in turn made them learn the materials better. Were this mechanism operating, L2 could be acting as a desirable difficulty that promotes learning [121].

Interestingly, this compensatory mechanism did not work to completely compensate for the difficulty of the materials; concrete words and related lists were still better recalled than abstract words and unrelated lists. Thus, although participants' JOLs were sensitive to the objective difficulty of the materials, they did not spontaneously use this knowledge to compensate for abstract and unrelated words and achieve the same level of learning as with the easier

materials. Note, however, that this pattern is in agreement with previous research in which participants did not compensate for the difficulty of the material even though the difficulty was perceived [e.g., 103]. It is possible that the mental representations for abstract and unrelated words, although objectively more difficult to encode and retrieve from memory than concrete or related words [65, 106, 122], might not provide a sufficient level of awareness to induce participants to engage in *control* strategies for compensation.

The fact that our participants were able to compensate for language difficulty but not for the difficulty of the materials may have to do with the differential distinctiveness of the cues for learning monitoring. Thus, it has been suggested [e.g., 43, 53] that the cues introduced in the materials and used by the learners to infer their degree of learning might be more or less distinctive, diagnostic, or informative. Therefore, it is possible that language may have been more distinctive for our participants than other features of the word, and they engaged regulatory learning strategies to compensate for the perceived difficulty of L2 contexts. Note that our participants, although proficient in their L2 language, were unbalanced and, therefore, might be highly aware of the difficulties associated with their less-proficient knowledge of L2. However, this explanation is speculative at the moment, since participants' JOLs indicated sensitivity to the difficulty of abstract and unrelated words, and we do not have direct comparisons of the participants' degree of awareness of the language cues. In the three experiments, language was manipulated between blocks to provide steady linguistic contexts (L1 vs. L2) where participants could use their metacognitive processes during learning, and therefore, it is not possible to assess the extent to which participants use language as a cue to assess their learning and control processes. Further research should address this question.

It is also important to note that our participants were selected because they were proficient late learners with an unbalanced use of their two languages. We thought this is crucial, since it mimicked the many situations in which proficient late L2 learners are required to use their second language, but it would be very relevant to investigate if lower levels of proficiency undermine learning monitoring and control. Note that, in all experiments, we also introduced proficiency as a covariate, and it did not have any effect in varying our results. However, our participants were selected to be very homogeneous in their languages, and it is possible that larger individual variations might modulate the results. How these effects differ depending on the linguistic characteristics, experience, and background of the participants (e.g., exposure, contexts and frequency of use) would be an intriguing avenue to explore in the future.

Two additional points have to do with the memory task and the learning strategies. As mentioned in the introduction, L2 (dis)advantages in memory are more evident in recall than in recognition tests. Hence, our pattern of results might have been different if we had included a recall test instead of a recognition one. The fact that tests in academic settings tend to include both formats make it relevant to assess and compare L2 memory by using both of them. Future studies would need to address this issue.

Finally, with regard to the learning strategies used by participants in the study phase, we found some commonalities and differences across experiment. Grouping words by their semantic meaning (86.1%), creating mental images (68.8%) and words rehearsal (76%) were the most prevalent strategies in experiments 1, 2 and 3 respectively. Interestingly, all these strategies reflect prevalence of semantic processing across the three experiments, although we observed subtle differences among them. The underlying reason that modulates such differences is not evident and further research should also address this issue. More importantly, this research should also address whether language related differences modulate the use of this strategies.

In sum, our results suggest that the complexity and characteristics of studied material might be a central aspect of the monitoring and control of learning in L2. Perceptual

manipulations might not have an impact on the semantic access of the word nor its mental representation and might not interfere in how people monitor their learning. Concrete and abstract word lists, along with related and unrelated word lists, required deeper processing, and language played a role in the monitoring and learning itself. Whether learning in L2 could be considered a desirable difficulty is a question that remains to be answered. The evidence presented here leans towards the idea of L2 learning leading to lower expectations but ultimately resulting in greater performance. The next step might be exploring this effect with increasingly complex materials and various L2 proficiency levels.

## Supporting information

**S1 File.**
(DOCX)

**S2 File.**
(XLSX)

## Author Contributions

**Conceptualization:** Marta Reyes, Mª Julia Morales, Mª Teresa Bajo.

**Data curation:** Marta Reyes, Mª Julia Morales.

**Formal analysis:** Marta Reyes.

**Funding acquisition:** Marta Reyes, Mª Teresa Bajo.

**Investigation:** Marta Reyes, Mª Julia Morales, Mª Teresa Bajo.

**Methodology:** Marta Reyes, Mª Julia Morales, Mª Teresa Bajo.

**Project administration:** Marta Reyes, Mª Julia Morales, Mª Teresa Bajo.

**Supervision:** Marta Reyes, Mª Julia Morales, Mª Teresa Bajo.

**Writing – original draft:** Marta Reyes.

**Writing – review & editing:** Marta Reyes, Mª Julia Morales, Mª Teresa Bajo.

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
