## [Decision Letter · Decision Letter 0]

8 Mar 2023

PONE-D-22-31354Judgments of learning in bilinguals: Does studying in a L2 hinder learning monitoring?PLOS ONE

Dear Dr. Reyes Sánchez:

Thank you for submitting your manuscript to PLOS ONE. After careful consideration, we feel that it has merit but does not fully meet PLOS ONE’s publication criteria as it currently stands. Therefore, we invite you to submit a revised version of the manuscript that addresses the points raised during the review process.

ACADEMIC EDITOR: 

Manuscript ID PONE-D-22-31354 entitled "Judgments of learning in bilinguals: Does studying in a L2 hinder learning monitoring?" which you submitted to Plos One, has been reviewed.  The comments of two reviewers are very detailed.

Both are positive and have recommended publication, but also suggest some revisions to your manuscript. I find myself in agreement with the review's general stance; therefore, I invite you to respond to the reviewer's comments and revise your manuscript before it can be considered for publication.

Journal Requirements:

We look forward to receiving your revised manuscript.

Kind regards,

Montserrat Comesaña Vila

Academic Editor

PLOS ONE

Journal Requirements:

Reviewers' comments:

Reviewer's Responses to Questions

**Comments to the Author**

1. Is the manuscript technically sound, and do the data support the conclusions?

Reviewer #1: Yes

Reviewer #2: Yes

2. Has the statistical analysis been performed appropriately and rigorously? 

Reviewer #1: Yes

Reviewer #2: Yes

3. Have the authors made all data underlying the findings in their manuscript fully available?

Reviewer #1: Yes

Reviewer #2: Yes

4. Is the manuscript presented in an intelligible fashion and written in standard English?

Reviewer #1: Yes

Reviewer #2: Yes

5. Review Comments to the Author

Reviewer #1: The manuscript presents three experiments to investigate the effects on recognition memory from studying lists of words in L1 and L2 and, more specifically, to explore whether the interplay between monitoring and control (metamemory processes) changes as a function of the language involved. Experiment 1 explored font type, Experiment examined concreteness, and Experiment 3 relatedness affected judgments of learning (JOLs) and memory performance in both L1 and L2. Results showed that people could monitor their learning in both L1 and L2, even though they judged L2 learning as less retrievable than L1. Interestingly, the self-perceived difficulty did not hinder learning, and people recognized L2 materials as well or better than L1 materials.

The paper leads with a very actual topic, and the results are interesting to be published. However, some concerns and suggestions need to be clarified by the authors for the paper to be considered for publication.

Introduction

The introduction is well-written and shows a clear presentation of arguments and support for the conceptual organization of the experiments. The authors present several relevant references associated with the main topic of the manuscript, namely the role of L1 and L2 representation on several tasks and the processes involved in metacognition.

Comment#1: at page 6, line 137, the authors said that “Generally, JOLs tend to be quite accurate in predicting recall performance (e.g., concrete and related words receive higher JOLs and are indeed better remembered). This sentence can induce the reader to think that the JOL assessment is a general procedure based on a “full list evaluation”. In fact, JOLs are an item-specific assessment, and the accuracy in predicting recall is not such high as the sentence suggests.

Comment#2: at page 7, line 165, the authors said that “Note that these manipulations are not equally predictive of learning success since, intrinsically, perceptual manipulation does not necessarily imply increased difficulty of the material, whereas concreteness and categorical relations have been shown to be more memorable than abstract and unrelated words.” There should be added an explanation regarding the relationship that this sentence turns implicit about the difficulty of the material (On what? Encoding? Retrieval? Monitoring?), higher memorability and the materials that were used in the three experiments of the manuscript. On page 8, line 175, the authors repeat the idea of “increase or decrease the difficulty of the material” by clarifying how such difficulty is operationalized. Please add some clarification.

Experiment 1 (perceptual manipulation)

Comment#3: at page 9, line 200, the authors affirm that “…font type was the only systematic cue on which to base JOLs and language acted as the contextual setting of learning.” Please add some short clarification. I also wonder why the authors did not choose font size as a perceptual manipulation since the results clearly indicate an effect on JOLs.

Comment#4: at page 10, line 345, there’s a reference regarding the completion by the participants of a metacognitive questionnaire to assess the strategies used during the procedure. There is no mention of these results in the manuscript, and we should be elucidated on the reasons for such omission.

Comment#5: at page 11, line 247, can the percentage of degraded grey be quantified? This is essential information due to the specific manipulation of this experiment and to turn accurate the replicability of the procedure.

Comment#6: at page 11, line 256, the author refers that “For each block, the list comprised 44 words, with the first and last two words serving as the primacy and recency buffers and the remaining 40 as targets. There were two lists (list A and list B) of 20 words in each language.” I thought that the language of the lists was presented in blocks, but in the sentence above, it seems that each list was composed of words in both languages. This needs some clarification of even a figure to explain it.

Comment#7: concerning the control for parameters that can influence word retrievability, please confirm that imageability and concreteness were not considered and why? I suppose such data is unavailable for all the words selected for the lists, but I need some clarification.

Comment#8: page 11, line 270: which kind of distractor task was implemented? A verbal one? A visual or spatial one?

Comment#9: page 12, line 272: it is unclear in the manuscript if, at the recognition test, the words presented as targets for some participants were given as fillers to others. This is a crucial control to avoid random effects of item-specific recognition on such tasks.

Comment#10: Table 2S in Supplementary Materials presented Misses and Omissions to the Target words. I miss the reason why. The Signal Detection Theory only considers two results to targets: hits and misses. In other words: what is the difference the authors are making between “misses” and “omissions”?

Comment#11: page 16, line 364: the authors refer to Magreehan’s paper to justify the absence of font type effect on Experiment 1 as a consequence of the presence of other cues available. Which are the other available cues, considering that the language was presented to the participants in blocks?

Comment#12: page 16, line 372: indeed, the effect of the order is very interesting. Making JOLs in the 2nd place to one of the languages allows the participants to calibrate (by comparison) the judgements. I wonder if a GK gamma correlation by block will not shed light on the capacity to judge memory retrieval.

Experiment 2 (lexical-semantic manipulation)

Comment#13: page 17, line 349: please clarify what you mean by online experiment. It means that the was a videoconference (e.g., zoom, teams, etc.), and the participants responded remotely? The same comment can be applied to Experiment 3.

Comment#14: page 18, line 426: are the values for concreteness and abstractness on L1 and L2 statistically equal? Clarification is needed, as also the presentation of the values in the Supplementary Materials. Otherwise, the comparison between languages could be biased and difficult to understand. On the other hand, a mean of 3.8 (SD = 0.7) on a 7-point scale as a reference to select the “abstract words” is arguable. Several words are considered abstract, but they have values higher than 4 (e.g., polvo = 4.54). Did I miss the rationale for the word selection of this experiment?

Comment#15: On page 20, line 466, please clarify what you mean by a d’ of .50 close to the chance level. D’ is a value based on hits and false alarm proportions, and the value of .50 can be obtained with a hit proportion of .80 (false alarm of .63), that is not a chance level performance.

Experiment 3 (semantic-relational manipulation)

Comment#16: On Experiment 3 materials and procedure, I missed the rationale for the JOLs to be implemented after the study phase of each list. The reference of Matvey et al. (2006), namely Experiment 2 of this paper, suggests a procedure that was not followed in this Experiment.

Comment#17: It seems that the distractors used on the recognition memory task were not the “same nature” as the targets. Exploring file 2 of the Supplementary Material, there are no categories used as fillers. Is there any reason not to control this aspect? Again, it seems also that the distractors presented to one participant were not used as targets (counterbalanced) to another participant.

Comment#18: Please clarify if, during the test phase (recognition memory), the words of the lists were presented in blocks or randomized.

Minor aspects

#1 On the Supplementary Materials (file 1), Table 1S is named S1.

#2 On the Supplementary Materials (file 1), Table 1S, how can a Mean of 0.00 have an SD of 0.32? If this is a question of decimals places, a note should be added to explain it.

#3 There is no Table 1 in the manuscript. The first table is named “Table 2”.

#4 I wonder if, in Table 1, a change in the first two columns (starting with “Language” and then “Block Order”) will not make more readable the “effects” on JOLs and d’ prime.

#5 On page 15, line 344, the reference is in a different format.

#6 On page 15, line 361, the expression “… difficult to study the words …” does not fit with the paper's aim and the JOLs task.

#7 page 20, line 468, instead of “recalled,” you should use “recognized”.

#8 page 23, line 536: the reference to Table 1A is incorrect.

Reviewer #2: This study examines metamemory processes in bilinguals, relying on the judgments of learning measure (JOL). Participants studied lists of Spanish (L1) and English (L2) words and gave JOL. After a distraction period, they did a recognition task. The study includes 3 experiments, each one with a different manipulation as a cue for the metamemory judgment: Font type in Experiment 1, concreteness in Experiment 2 and semantic relatedness in Experiment 3. The results show that bilinguals can monitor their learning in both L1 and L2 and that they recognize words equally in L1 and L2 (and in some cases, even better in L2), although they judge learning L2 words as more difficult than learning L1 words. This is an interesting and well conducted study. It addresses a relevant and timely topic, considering current educational practices involving the teaching of content in different languages. The manuscript is clear and well written. I have several concerns, however, that should be addressed before the paper is accepted. I list them below:

-I wonder why the authors decided to use a recognition task instead of a free recall task. Vander Beken and Brysbaert (2018) found an advantage in L1 with respect to L2 in recall, but not in recognition. The literature comparing word recall and recognition in L2 vs L1 should be included in the introduction and discussion. The present results need to be related to this literature. Do the authors believe that their results would have been different had they used a free recall task?

-In Experiment 1, there were no effects of font type either in L1 or in L2 in JOLs. The manipulation did not affect recognition rates either. I wonder why the authors chose this manipulation, considering that, as they state in the introduction, “the evidence for the effect of perceptual manipulations on JOLs and memory is mixed” (page 8).

-May the authors explain the nature of the distracting task?

-Experiment 2 was focused on concreteness. Reference 56 of Romani and co-workers is included as evidence of concreteness effects. This reference is about the effects of concreteness in short memory, but the paradigm used in this study is not a short-term memory paradigm. Please, include some more appropriate reference about concreteness effects in long-term memory.

-The authors need to clarify whether they consider concreteness a lexical or a semantic variable. At the top of page 17, they state that concreteness is a lexical manipulation, but in the following paragraph concreteness is related to conceptual processing. This is confusing.

-Page 17: It is stated that associations between words and their meanings are weaker in L2 than in L1, according to the RHM of Kroll and co-workers. The model proposes that this is the case for beginner bilinguals. However, bilinguals in this study are not in the initial stages of L2 acquisition. Please, clarify this issue.

-Looking at the supplementary materials of Experiment 2, it appears that, overall, the Spanish words have higher concreteness values than the English words. Can the authors confirm this? Which was the criterion to decide that a word was concrete/abstract? Were concreteness values for Spanish words obtained from a normative database in Spanish?

Minor: In the description of the recognition data, it is said several times that participants "recalled".....words. But this is a recognition test, participants did not recall anything, they recognized the words. Please check it.

6. PLOS authors have the option to publish the peer review history of their article (what does this mean?). If published, this will include your full peer review and any attached files.

Reviewer #1: No

Reviewer #2: No

---

## [Author Response · Author response to Decision Letter 0]

18 Apr 2023

Granada, April 18, 2023

Dear Dr. Montserrat Comesaña Vila,

We sincerely appreciate the opportunity to submit a revised version of our paper PONE-D-22-31354 entitled "Judgments of learning in bilinguals: Does studying in a L2 hinder learning monitoring?". We have reviewed the manuscript to address the concerns raised by the reviewers as well as your comments and suggestions. Changes in the manuscript are highlighted in red and below, we list our responses to their comments in blue font. We appreciate all comments and suggestions. We strongly believe that the changes have significantly improved the manuscript. 

Sincerely,

Marta Reyes (on behalf of all co-authors).

Journal Requirements: 

Authors: We apologize for not meeting the style requirements entirely in our first submission. We have now updated the format accordingly. 

Authors: Following the editor’s suggestion, data have been uploaded to the Open Science Framework (OSF) public repository as separate .csv files for each experiment, as well as codebooks as .txt files with the explanation of each variable. Please, find the data in the following OSF identifier: https://osf.io/e7qyn/?view_only=1e2034b6d38641bab4a460172a530997

Authors: We apologize for having missed this information in our first submission. We have now included this information in a subsection at the end of the manuscript. 

Reviewers’ comments: 

Reviewer #1: The manuscript presents three experiments to investigate the effects on recognition memory from studying lists of words in L1 and L2 and, more specifically, to explore whether the interplay between monitoring and control (metamemory processes) changes as a function of the language involved. Experiment 1 explored font type, Experiment examined concreteness, and Experiment 3 relatedness affected judgments of learning (JOLs) and memory performance in both L1 and L2. Results showed that people could monitor their learning in both L1 and L2, even though they judged L2 learning as less retrievable than L1. Interestingly, the self-perceived difficulty did not hinder learning, and people recognized L2 materials as well or better than L1 materials.

The paper leads with a very actual topic, and the results are interesting to be published. However, some concerns and suggestions need to be clarified by the authors for the paper to be considered for publication.

Introduction

The introduction is well-written and shows a clear presentation of arguments and support for the conceptual organization of the experiments. The authors present several relevant references associated with the main topic of the manuscript, namely the role of L1 and L2 representation on several tasks and the processes involved in metacognition.

Comment#1: at page 6, line 137, the authors said that “Generally, JOLs tend to be quite accurate in predicting recall performance (e.g., concrete and related words receive higher JOLs and are indeed better remembered). This sentence can induce the reader to think that the JOL assessment is a general procedure based on a “full list evaluation”. In fact, JOLs are an item-specific assessment, and the accuracy in predicting recall is not such high as the sentence suggests.

Authors: We appreciate this comment which allows us to clarify some features of the JOL procedure. In lines 121-123 (pg. 5-6), we clarify that the JOLs can be based on a full list or item-specific evaluation, and this depends on the purpose of the study and the instructions given. 

In addition, in line 140-141 (pg. 6), we have softened the sentence regarding the accuracy of JOLS in predicting subsequent recall. Yet, although JOLs tend to be quite accurate and correlate with actual memory performance for certain cues (e.g., concreteness and relatedness, Hertzog et al., 2003; Undorf & Erdfelder, 2015; Witherby & Tauber, 2017), they are sensitive to manipulations and context. We have introduced the word “usually” to illustrate that there might be conditions where people are not that accurate and we have also added some references. 

Comment#2: at page 7, line 165, the authors said that “Note that these manipulations are not equally predictive of learning success since, intrinsically, perceptual manipulation does not necessarily imply increased difficulty of the material, whereas concreteness and categorical relations have been shown to be more memorable than abstract and unrelated words.” There should be added an explanation regarding the relationship that this sentence turns implicit about the difficulty of the material (On what? Encoding? Retrieval? Monitoring?), higher memorability and the materials that were used in the three experiments of the manuscript. 

Authors: Thanks for pointing this out, We have now included a sentence to specify what we mean (see lines 184-186, pg. 8) and make it explicit that difficulty refers to the impact that these manipulations may have in encoding and retrieval processes. 

On page 8, line 175, the authors repeat the idea of “increase or decrease the difficulty of the material” by clarifying how such difficulty is operationalized. Please add some clarification.

Authors: An example of the two manipulations used in our experiments has been included to clarify what we meant by “increase or decrease the difficulty of the material” (see lines 193-194, pg. 8).

Experiment 1 (perceptual manipulation)

Comment#3: at page 9, line 200, the authors affirm that “…font type was the only systematic cue on which to base JOLs and language acted as the contextual setting of learning.” Please add some short clarification. I also wonder why the authors did not choose font size as a perceptual manipulation since the results clearly indicate an effect on JOLs.

Authors: We have added some clarification in lines 220-222, pg. 9-10, with reference to the language acting as the contextual setting of learning.

With regard to our perceptual manipulation choice, we thought of the font-size effect when designing the experiment, but the main reason to discard font-size as the target manipulation is that we planned to follow up this experiment by a second one where eye-tracking would be used to analyze pupillometry data. Since variations in font size may cause differences in pupillometry and act as a confound, we decided to manipulate font type. However, we were never able to run the eye-tracking experiment because the laboratory shut down due to the COVID-19 pandemic, which lead us to adapt our initial research project. 

Comment#4: at page 10, line 345, there’s a reference regarding the completion by the participants of a metacognitive questionnaire to assess the strategies used during the procedure. There is no mention of these results in the manuscript, and we should be elucidated on the reasons for such omission.

Authors: Information on the questionnaires has been now included in the results section (see lines 389-391, 542-545, 689-691, pg. 17, 23 and 30 respectively) and in the general discussion of the manuscript (see lines 812-820, pg. 34). For experiment 1, participants reported grouping words by their semantic meaning (86.1%), creating mental images (69.4%) and words rehearsal (52.8%) as the strategies most used. In experiment 2, participants reported creating mental images (68.8%), words rehearsal (59.4%), grouping words by their semantic meaning (59.4%), and relating words to personal experiences (56.3%) as the strategies most used in the study phase. In experiment 3, participants reported words rehearsal (76%), grouping words by their semantic meaning (68.3%), and creating mental images (56.1%) as the strategies most used in the study phase.

We had not included this information in the previous version of the article because we initially intended this questionnaire to explore the metacognitive strategies used in the study phase and compared them between languages. Nevertheless, we programmed it in such a way that participants only responded to the frequency of the strategies without specifying the language in which they were making used of them. Hence, we thought it was not informative, but we now understand that it provides some information of the changes in strategies depending on the metacognitive cues provided by the material in the different experiment that might also be useful for the readers. Thanks for the suggestion. 

Comment#5: at page 11, line 247, can the percentage of degraded grey be quantified? This is essential information due to the specific manipulation of this experiment and to turn accurate the replicability of the procedure.

Authors: We appreciate this comment. These details have been now included in the manuscript (see lines 270-272, pg. 12). 

Comment#6: at page 11, line 256, the author refers that “For each block, the list comprised 44 words, with the first and last two words serving as the primacy and recency buffers and the remaining 40 as targets. There were two lists (list A and list B) of 20 words in each language.” I thought that the language of the lists was presented in blocks, but in the sentence above, it seems that each list was composed of words in both languages. This needs some clarification of even a figure to explain it.

Authors: We have now introduced some sentences in the text to clarify the blocking and counterbalancing procedure. We explain that language was blocked such that in each study/recognition phase words appeared either in L1 or L2. List A and list B were used to randomly assign words to either font type. Within each block (L1 or L2), participants studied 40 words (after removing primacy and recency buffers), half of them in an easy-to-read font and half of them in a difficult-to-read font type, which was counterbalanced across participants (see line 282-284, 286-288, pg. 12).

Comment#7: concerning the control for parameters that can influence word retrievability, please confirm that imageability and concreteness were not considered and why? I suppose such data is unavailable for all the words selected for the lists, but I need some clarification.

Authors: We have checked the manuscript to make sure that the text clearly states the parameters of the materials that we controlled. In experiment 1, the study lists contained words matched for estimated frequency, number of letters (length), number of phonological neighbors, and number of orthographic neighbors. With regard to the recognition test, new words included were matched with the studied words for mean estimated frequency and mean number of letters, only. Although, we considered to control also for concreteness, it was not possible given the countless missing values for concreteness ratings for the words selected. 

Comment#8: page 11, line 270: which kind of distractor task was implemented? A verbal one? A visual or spatial one?

Authors: We used a short version of the AX- Continuous Performance Task (AX- CPT; Morales et al., 2013), which is a standard cognitive control task with minimum verbal load that we usually include in most experiments with bilinguals at our research group. We have now mentioned the AX-CPT in the manuscript (see lines 301-302, pg. 13). 

Comment#9: page 12, line 272: it is unclear in the manuscript if, at the recognition test, the words presented as targets for some participants were given as fillers to others. This is a crucial control to avoid random effects of item-specific recognition on such tasks.

Authors: We have now clarified in the text (see lines 305-307, 309-310, pg. 13) that targets and fillers were not counterbalanced across participants, but that they were matched in frequency and number of letters so that any possible effect that may arise would not be explained by those psycholinguistic parameters (see Lanska et al., 2014; Wehr & Wippich, 2004; Yue et al., 2013 for a similar procedure).

Comment#10: Table 2S in Supplementary Materials presented Misses and Omissions to the Target words. I miss the reason why. The Signal Detection Theory only considers two results to targets: hits and misses. In other words: what is the difference the authors are making between “misses” and “omissions”?

Authors: We agree that following the Signal Detection Theory, table 2S should only present hits (responding YES to previous study items), false alarms (responding YES to new items), misses (responding NO to previous study items) and correct rejections (responding NO to new items). However, we also included the percentage of omissions to show the absence of response (people not responding anything) as we thought it could be informative. However, since the label might lead to confusion, we have now changed the label “omission” to “no response” in every table. Thanks for the comment.

Comment#11: page 16, line 364: the authors refer to Magreehan’s paper to justify the absence of font type effect on Experiment 1 as a consequence of the presence of other cues available. Which are the other available cues, considering that the language was presented to the participants in blocks?

Authors: This comment is more than relevant. We have now elaborated our arguments in lines 402-408, pg. 17. It is true that within each block, participants only had the perceptual manipulation as a varying cue across items. However, participants had been fully informed of the procedure at the beginning of the experiment. They already knew that they were going to study words in two languages and that within each language block, words could appear in two different font types. They were instructed to judge their learning based on the difficulty perceived with all the information available. Despite language being a blocked variable, participants were immersed in a setting that also provided valuable information to inform JOLs. Thus, we believe that JOLs were based on both, the systematic cue of font type and the blocked cue of language. In fact, across the three experiments results show a tendency towards L1 and L2 receiving different JOLs values. 

Comment#12: page 16, line 372: indeed, the effect of the order is very interesting. Making JOLs in the 2nd place to one of the languages allows the participants to calibrate (by comparison) the judgements. I wonder if a GK gamma correlation by block will not shed light on the capacity to judge memory retrieval.

Authors: As mentioned in lines 370-379 pg. 16, we calculated one gamma correlation for each participant in each of the four conditions of interest (L1 easy-to-read, L1 difficult-to-read, L2 easy-to-read, L2 difficult-to-read). As block order was manipulated between subjects, we could not take it into consideration when computing the gamma correlation for each participant. Nevertheless, we included it in the ANOVA as a between-subject factor to examine whether the GK gamma correlations differed across conditions. The main effect of block order was not significant F(1, 16) = 0.04, p = .85, ηp2 = .002). Neither was any of the other interactions (all ps > .05), see Table 2). We have now explicitly mentioned that the interaction with blocked order was not significant (lines 385-387, pg.16) and also included this information in the discussion (lines 419-420, pg. 18).

Experiment 2 (lexical-semantic manipulation)

Comment#13: page 17, line 349: please clarify what you mean by online experiment. It means that the was a videoconference (e.g., zoom, teams, etc.), and the participants responded remotely? The same comment can be applied to Experiment 3.

Authors: We have now clarified this issue in the manuscript (see lines 463-468, 599, pg. 20, 26). Experiments 2 and 3 were conducted remotely. Participants accessed the link to the experimental procedure and follow all the instructions on their own. Recent research supports the validity and precision of experiments run online (Anwyl-Irvine et al., 2020, 2021; Gagné & Franzen, 2023). 

Comment#14: page 18, line 426: are the values for concreteness and abstractness on L1 and L2 statistically equal? Clarification is needed, as also the presentation of the values in the Supplementary Materials. Otherwise, the comparison between languages could be biased and difficult to understand. On the other hand, a mean of 3.8 (SD = 0.7) on a 7-point scale as a reference to select the “abstract words” is arguable. Several words are considered abstract, but they have values higher than 4 (e.g., polvo = 4.54). Did I miss the rationale for the word selection of this experiment?

Authors: Thanks for noticing that this was ambiguous. We have now clarified this issue in lines 486-493, pg. 21 and in the Supporting Information file ( S5 Table). We used two different language specific norms to select the words. Thus, concreteness ratings for English words were based on Brysbaert et al., (2014) using a 5-point scale, whereas values for Spanish words were based on LEXESP (Sebastián et al., 2000) using a 7-point scale. Thus, the descriptive statistics are in different scales (see the table below that we have also included now in the Supporting Information). However, the criteria to consider a word abstract or concrete was equivalent for both data set. We calculated the mean concreteness for each language, and words with ratings above the means in both languages were considered concrete, whereas words with values below the means were considered abstract.

 Spanish – L1 English – L2

Mean (SD) concreteness rating Concrete 5.78 (0.5) 4.56 (0.4)

 Abstract 3.76 (0.69) 2.57 (0.67)

 Total 4.7 (1.18) 3.64 (1.13)

Min. concreteness rating Concrete 4.8 3.64

 Abstract 2.22 1.25 

Max. concreteness rating Concrete 6.66 5

 Abstract 4.79 3.54

Comment#15: On page 20, line 466, please clarify what you mean by a d’ of .50 close to the chance level. D’ is a value based on hits and false alarm proportions, and the value of .50 can be obtained with a hit proportion of .80 (false alarm of .63), that is not a chance level performance.

Authors: Thanks for the comment, you are right, d’ = .50 does not necessarily mean “chance level”, so we have taken this expression out of the paper (see lines 525, 667, pg. 22, 28). In our case, all participants with d’= 0.50 showed a pattern of hits and false alarms indicating chance-level performance. However, the expression is not appropriate when describing our d’ criteria. Thanks again for the comment. 

Experiment 3 (semantic-relational manipulation)

Comment#16: On Experiment 3 materials and procedure, I missed the rationale for the JOLs to be implemented after the study phase of each list. The reference of Matvey et al. (2006), namely Experiment 2 of this paper, suggests a procedure that was not followed in this Experiment.

Authors: Thanks for the comment, we have now clarified that we used an adapted procedure, modeled after Matvey et al., (2006) (see lines 610-618, pg. 26). The main difference is that JOLs were provided after each list and not after each item. We had two reasons for this change: 1) the manipulation affected the complete list (related and unrelated lists), differently from experiments 1 and 2 were the manipulation affected specific words within the list (e.g., concrete words vs. abstract words), hence it was possible to assess the difficulty of the list as a whole; 2) collecting JOLs after each list made the procedure less cumbersome for the participants. 

Comment#17: It seems that the distractors used on the recognition memory task were not the “same nature” as the targets. Exploring file 2 of the Supplementary Material, there are no categories used as fillers. Is there any reason not to control this aspect? Again, it seems also that the distractors presented to one participant were not used as targets (counterbalanced) to another participant.

Authors: We have now clarified that it was not possible to include semantic categories among the new words in the recognition tests (see lines 638- 642, pg. 27). This was not possible given our selection procedure. We selected a wide pool of words in English, translated them into Spanish, matched them between languages and counterbalanced the category words across participants. Hence, after controlling for estimated frequency and number of letters, many semantic categories were discarded and we did not have enough categories to use as fillers. As in the previous experiments, all unrelated words were matched in frequency and number of letters and, for the sake of programming, we selected half of them to comprised the study phase and the other half to serve as fillers for the recognition test. Despite this limitation, we think that our design is all right for our purposes since the main comparison was between languages and the nature of the new-words in the recognition test were equal for the two language conditions. In our view, controlling for language was paramount to rule out any possible confounding results.

Comment#18: Please clarify if, during the test phase (recognition memory), the words of the lists were presented in blocks or randomized.

Authors: Words appeared randomly regardless of the condition (words grouped into semantic related categories and unrelated words). This has been clarified in lines 642-644, pg. 27.

Minor aspects

Authors: We really appreciate all minor aspects raised regarding typo mistakes and suggestions to improve the format. They have been implemented (comments number 1, 3, 5, 7 and 8)

#1 On the Supplementary Materials (file 1), Table 1S is named S1.

#2 On the Supplementary Materials (file 1), Table 1S, how can a Mean of 0.00 have an SD of 0.32? If this is a question of decimals places, a note should be added to explain it.

Authors: Indeed, it is a matter of decimals places. A note has been included to specified this issue and provide the full mean value.

#3 There is no Table 1 in the manuscript. The first table is named “Table 2”.

#4 I wonder if, in Table 1, a change in the first two columns (starting with “Language” and then “Block Order”) will not make more readable the “effects” on JOLs and d’ prime.

Authors: We thanks the suggestion and agree that it facilitates the readability of the table. We have changed the first two columns of every table (not just table 1) so as to be consistent. 

#5 On page 15, line 344, the reference is in a different format.

#6 On page 15, line 361, the expression “… difficult to study the words …” does not fit with the paper's aim and the JOLs task.

Authors: We agree this phrase was out of the scope of the task and we have paraphrase it stating that “Participants predicted similar memory performance for words in a difficult-to-read and easy-to-read font.” (See line 399, pg. 17).

#7 page 20, line 468, instead of “recalled,” you should use “recognized”.

#8 page 23, line 536: the reference to Table 1A is incorrect.

Reviewer #2: This study examines metamemory processes in bilinguals, relying on the judgments of learning measure (JOL). Participants studied lists of Spanish (L1) and English (L2) words and gave JOL. After a distraction period, they did a recognition task. The study includes 3 experiments, each one with a different manipulation as a cue for the metamemory judgment: Font type in Experiment 1, concreteness in Experiment 2 and semantic relatedness in Experiment 3. The results show that bilinguals can monitor their learning in both L1 and L2 and that they recognize words equally in L1 and L2 (and in some cases, even better in L2), although they judge learning L2 words as more difficult than learning L1 words. This is an interesting and well conducted study. It addresses a relevant and timely topic, considering current educational practices involving the teaching of content in different languages. The manuscript is clear and well written. I have several concerns, however, that should be addressed before the paper is accepted. I list them below:

-I wonder why the authors decided to use a recognition task instead of a free recall task. Vander Beken and Brysbaert (2018) found an advantage in L1 with respect to L2 in recall, but not in recognition. The literature comparing word recall and recognition in L2 vs L1 should be included in the introduction and discussion. The present results need to be related to this literature. Do the authors believe that their results would have been different had they used a free recall task?

Authors: Thanks for the comment. We agree that those studies are relevant to understand the scope of our results and have now included them in the introduction (lines, 160-173, pg. 7-8), and the discussion (lines 807-812, pg. 32). 

Indeed, and according to the literature, we believe that the pattern of results might have been different if we had included a free recall test instead of a recognition test. When designing the experiments, we did consider employing recall tests because the truth is that tests in academic settings tend to include both formats and, therefore, both are of interest. However, in the current paper we decided to employ recognition tests to assess memory performance in order to avoid confounding effects with writing complexity. We have also introduced some sentences in the discussion acknowledging that if we had used free recall as the final test the results might have been different and that future studies should address this issue (lines 810-812, pg. 32).

-In Experiment 1, there were no effects of font type either in L1 or in L2 in JOLs. The manipulation did not affect recognition rates either. I wonder why the authors chose this manipulation, considering that, as they state in the introduction, “the evidence for the effect of perceptual manipulations on JOLs and memory is mixed” (page 8).

Authors: This comment was also raised by reviewer 1 and suggested that we might have considered font size as a better manipulation. The truth is that we thought of it when designing the experiment, but main reason to discard font-size as the target manipulation is that we planned to follow up this experiment by a second one where eye-tracking would be used to analyze pupillometry data. Since variations in font size may cause differences in pupillometry and act as a confound, we decided to manipulate font type. However, we were never able to run the eye-tracking experiment because the laboratory shut down due to the COVID-19 pandemic, which lead us to adapt our initial research project as the labs were not fully operative.

-May the authors explain the nature of the distracting task?

Authors: This was also raised by Reviewer 1, and we have included some details of the distractor task in the procedure (lines 301-302, pg. 13).

-Experiment 2 was focused on concreteness. Reference 56 of Romani and co-workers is included as evidence of concreteness effects. This reference is about the effects of concreteness in short memory, but the paradigm used in this study is not a short-term memory paradigm. Please, include some more appropriate reference about concreteness effects in long-term memory.

Authors: Thanks for noticing, you are right, we have now taken this reference out and included other references regarding the concreteness effect in recognition memory (Begg et al., 1989; De Groot & Keijzer, 2000; Paivio, 1991) (See line 435, pg. 19).

-The authors need to clarify whether they consider concreteness a lexical or a semantic variable. At the top of page 17, they state that concreteness is a lexical manipulation, but in the following paragraph concreteness is related to conceptual processing. This is confusing.

Authors: We really appreciate this comment as we lack consistency when using one term or the other, which was certainly causing confusion in the manuscript. We have now changed the label of the manipulations for experiment 2 (lexical-semantic) and experiment 3 (semantic-relational), and used them in a consistent manner along the text.

Concreteness is assumed to have a semantic component which affect to specific words. The ‘‘dual-coding theory’’ (Paivio, 1991), argues that access to visual properties (“images”) of concrete words facilitates the access to the meaning. In addition, concrete words have been shown to have more integrated representations than abstract words, but this semantic property refers to individual word representation, and this is different from the type of semantic relational information included in the materials of Experiment 3. That is why we used the terms lexical-semantic (experiment 2) and relational-semantics (Experiment 3). 

-Page 17: It is stated that associations between words and their meanings are weaker in L2 than in L1, according to the RHM of Kroll and co-workers. The model proposes that this is the case for beginner bilinguals. However, bilinguals in this study are not in the initial stages of L2 acquisition. Please, clarify this issue.

Authors: As far as we know, the Revised Hierarchical Model (Kroll & Stewart, 1994) is based on bilingual memory representation and its appliance is not restricted only to beginner L2 learners. We agree that the ease of accessing connections between L2 words and concepts changes dramatically as proficiency in L2 increases. However, our sample was unbalanced bilinguals with an intermediate level of L2 who may lack the sufficient proficiency as to overcome the asymmetry proposed in the RHM. Hence, we strongly believe that participants in our study had weaker associations between words and their meanings in L2. We have now clarified this point in lines 441, pg. 19.

-Looking at the supplementary materials of Experiment 2, it appears that, overall, the Spanish words have higher concreteness values than the English words. Can the authors confirm this? Which was the criterion to decide that a word was concrete/abstract? Were concreteness values for Spanish words obtained from a normative database in Spanish?

Authors: Thanks for the comment. This was also raised by reviewer 1 and clearly needed some explanation in the article (see lines 486-493, pg. 21). We used two different language specific norms to select the words. Thus, concreteness ratings for English words were based on Brysbaert et al., (2014) using a 5-point scale, whereas values for Spanish words were based on LEXESP (Sebastián et al., 2000) using a 7-point scale. Thus, the descriptive statistics are in different scales (see the table below, which we have also included now in the Supporting Information, 5S Table). However, the criteria to consider a word abstract or concrete was equivalent for both data set. We calculated the mean concreteness for each language, and words with ratings above the means in both languages were considered concrete, whereas words with values below the means were considered abstract.

 Spanish – L1 English – L2

Mean (SD) concreteness rating Concrete 5.78 (0.5) 4.56 (0.4)

 Abstract 3.76 (0.69) 2.57 (0.67)

 Total 4.7 (1.18) 3.64 (1.13)

Min. concreteness rating Concrete 4.8 3.64

 Abstract 2.22 1.25 

Max. concreteness rating Concrete 6.66 5

 Abstract 4.79 3.54

However, the criteria to consider a word abstract or concrete was the same in both data set: we split the pool in to halves by the mean. Words with ratings above the means in both languages were considered concrete and words with value below the means were considered abstract. We have introduced this explanation in lines 491-497, pg. 21.

Minor: In the description of the recognition data, it is said several times that participants "recalled".....words. But this is a recognition test, participants did not recall anything, they recognized the words. Please check it.

Authors: We really appreciate this minor aspect raised. We have now replaced the term “recalled”, which was inaccurate, with the term “recognized”. 

References

Anwyl-Irvine, A., Dalmaijer, E. S., Hodges, N., & Evershed, J. K. (2021). Realistic precision and accuracy of online experiment platforms, web browsers, and devices. Behavior Research Methods, 53(4), 1407–1425. https://doi.org/10.3758/s13428-020-01501-5

Anwyl-Irvine, A., Massonnié, J., Flitton, A., Kirkham, N., & Evershed, J. K. (2020). Gorilla in our midst: An online behavioral experiment builder. Behavior Research Methods, 52(1), 388–407. https://doi.org/10.3758/s13428-019-01237-x

Begg, I., Duft, S., Lalonde, P., Melnick, R., & Sanvito, J. (1989). Memory predictions are based on ease of processing. Journal of Memory and Language, 28(5), 610–632. https://www.unhcr.org/publications/manuals/4d9352319/unhcr-protection-training-manual-european-border-entry-officials-2-legal.html?query=excom 1989

Brysbaert, M., Warriner, A. B., & Kuperman, V. (2014). Concreteness ratings for 40 thousand generally known English word lemmas. Behavior Research Methods, 46(3), 904–911. https://doi.org/10.3758/s13428-013-0403-5

De Groot, A. M. B., & Keijzer, R. (2000). What is hard to learn is easy to forget: The roles of word concreteness, cognate status, and word frequency in foreign-language vocabulary learning and forgetting. Language Learning, 50, 1–56.

Gagné, N., & Franzen, L. (2023). How to Run Behavioural Experiments Online: Best Practice Suggestions for Cognitive Psychology and Neuroscience. Swiss Psychology Open, 3(1), 1. https://doi.org/10.5334/spo.34

Hertzog, C., Dunlosky, J., Emanuel Robinson, A., & Kidder, D. P. (2003). Encoding Fluency Is a Cue Used for Judgments About Learning. Journal of Experimental Psychology: Learning Memory and Cognition, 29(1), 22–34. https://doi.org/10.1037/0278-7393.29.1.22

Kroll, J. F., & Stewart, E. (1994). Category Interference in Translation and Picture Naming: Evidence for Asymmetric Connections between Bilingual Memory Representations. Journal of Memory and Language, 33, 149–174.

Lanska, M., Olds, J. M., & Westerman, D. L. (2014). Fluency effects in recognition memory: Are perceptual fluency and conceptual fluency interchangeable. Journal of Experimental Psychology: Learning Memory and Cognition, 40(1), 1–11. https://doi.org/10.1037/a0034309

Matvey, G., Dunlosky, J., & Schwartz, B. L. (2006). The effects of categorical relatedness on judgements of learning (JOLs). Memory, 14(2), 253–261. https://doi.org/10.1080/09658210500216844

Morales, J., Gómez-Ariza, C. J., & Bajo, M. T. (2013). Dual mechanisms of cognitive control in bilinguals and monolinguals. Journal of Cognitive Psychology, 25, 531–546.

Paivio, A. (1991). Dual coding theory: Retrospect and current status. Canadian Journal of Psychology, 45(255–287).

Sebastián, N., Martí, M. A., Carreiras, M. F., & Cuetos, F. (2000). LEXESP, léxico informatizado del español. Ediciones de la Universitat de Barcelona.

Undorf, M., & Erdfelder, E. (2015). The relatedness effect on judgments of learning: A closer look at the contribution of processing fluency. Memory and Cognition, 43(4), 647–658. https://doi.org/10.3758/s13421-014-0479-x

Wehr, T., & Wippich, W. (2004). Typography and color: effects of salience and fluency on conscious recollective experience. Psychological Research, 69(1–2), 138–146. https://doi.org/10.1007/s00426-003-0162-5

Witherby, A. E., & Tauber, S. K. (2017). The concreteness effect on judgments of learning: Evaluating the contributions of fluency and beliefs. Memory and Cognition, 45(4), 639–650. https://doi.org/10.3758/s13421-016-0681-0

Yue, C. L., Castel, A. D., & Bjork, R. A. (2013). When disfluency is -and is not- a desirable difficulty: The influence of typeface clarity on metacognitive judgments and memory. Memory & Cognition, 41, 229–241.

---

## [Decision Letter · Decision Letter 1]

18 May 2023

Judgments of learning in bilinguals: Does studying in a L2 hinder learning monitoring?

PONE-D-22-31354R1

Dear Dr. Reyes Sánchez,

We’re pleased to inform you that your manuscript has been judged scientifically suitable for publication and will be formally accepted for publication once it meets all outstanding technical requirements.

Kind regards,

Montserrat Comesaña Vila

Academic Editor

PLOS ONE

Reviewers' comments:

Reviewer's Responses to Questions

**Comments to the Author**

1. If the authors have adequately addressed your comments raised in a previous round of review and you feel that this manuscript is now acceptable for publication, you may indicate that here to bypass the “Comments to the Author” section, enter your conflict of interest statement in the “Confidential to Editor” section, and submit your "Accept" recommendation.

Reviewer #1: All comments have been addressed

Reviewer #2: (No Response)

2. Is the manuscript technically sound, and do the data support the conclusions?

Reviewer #1: Yes

Reviewer #2: Yes

3. Has the statistical analysis been performed appropriately and rigorously? 

Reviewer #1: Yes

Reviewer #2: Yes

4. Have the authors made all data underlying the findings in their manuscript fully available?

Reviewer #1: Yes

Reviewer #2: Yes

5. Is the manuscript presented in an intelligible fashion and written in standard English?

Reviewer #1: Yes

Reviewer #2: Yes

6. Review Comments to the Author

Reviewer #1: The authors made a considerable effort to adequately answer all my comments and suggestions in the first version of the manuscript. I wish to remark that this new version of the manuscript is more precise, and, as I mentioned before, it leads with a very actual topic, and the results are exciting and could be a reference to future studies in this field. Resuming, I recommend the publication of the paper in this present form.

Reviewer #2: (No Response)

7. PLOS authors have the option to publish the peer review history of their article (what does this mean?). If published, this will include your full peer review and any attached files.

Reviewer #1: **Yes: **Pedro B. Albuquerque

Reviewer #2: No

---

## [Editor Report · Acceptance letter]

24 Nov 2023

PONE-D-22-31354R1 

Judgments of learning in bilinguals: Does studying in a L2 hinder learning monitoring? 

Dear Dr. Reyes:

I'm pleased to inform you that your manuscript has been deemed suitable for publication in PLOS ONE. Congratulations! Your manuscript is now with our production department. 

Kind regards, 

on behalf of

Dr. Montserrat Comesaña Vila 

Academic Editor

PLOS ONE